# Pseudorabies Virus Glycoproteins E and B Application in Vaccine and Diagnosis Kit Development

**DOI:** 10.3390/vaccines12091078

**Published:** 2024-09-20

**Authors:** Sara Amanuel Bude, Zengjun Lu, Zhixun Zhao, Qiang Zhang

**Affiliations:** 1State Key Laboratory for Animal Disease Control and Prevention, College of Veterinary Medicine, Lanzhou Veterinary Research Institute, Chinese Academy of Agricultural Sciences, Lanzhou 730000, China; sara.amanuel@aau.edu.et (S.A.B.); luzengjun@caas.cn (Z.L.); 2College of Veterinary Medicine and Agriculture, Addis Ababa University, Bishoftu P.O. Box 34, Ethiopia

**Keywords:** glycoprotein B (gB), glycoprotein E (gE), pseudorabies virus (PRV), swine vaccine

## Abstract

**Background**: Pseudorabies virus (PRV) is a highly infectious pathogen that affects a wide range of mammals and imposes a significant economic burden on the global pig industry. The viral envelope of PRV contains several glycoproteins, including glycoprotein E (gE) and glycoprotein B (gB), which play critical roles in immune recognition, vaccine development, and diagnostic procedures. Mutations in these glycoproteins may enhance virulence, highlighting the need for updated vaccines. **Method**: This review examines the functions of PRV gE and gB in vaccine development and diagnostics, focusing on their roles in viral replication, immune system interaction, and pathogenicity. Additionally, we explore recent findings on the importance of gE deletion in attenuated vaccines and the potential of gB to induce immunity. **Results**: Glycoprotein E (gE) is crucial for the virus’s axonal transport and nerve invasion, facilitating transmission to the central nervous system. Deletion of gE is a successful strategy in vaccine development, enhancing the immune response. Glycoprotein B (gB) plays a central role in viral replication and membrane fusion, aiding viral spread. Mutations in these glycoproteins may increase PRV virulence, complicating vaccine efficacy. **Conclusion**: With PRV glycoproteins being essential to both vaccine development and diagnostic approaches, future research should focus on enhancing these components to address emerging PRV variants. Updated vaccines and diagnostic tools are critical for combating new, more virulent strains of PRV.

## 1. Introduction

Pseudorabies virus (PRV), also known as Aujeszky’s disease virus, is a significant pathogen affecting pigs and other animals, leading to substantial economic losses and severe health issues [1]. PRV is an enveloped virus with a linear double-stranded DNA genome approximately 143 kb in size, containing at least 72 genes [2]. It belongs to the genus Varicella within the herpesvirus A subfamily of the herpesvirus family, and is also referred to as SuHV-1 (SuHV-1) [3,4].

The PRV envelope is embedded with numerous glycoproteins, each serving distinct functions [5,6,7,8,9,10,11]. Key glycoproteins such as gB, gD, gH, and gL are crucial for the virus’s entry into host cells [12,13,14,15,16]. Other glycoproteins, including gC, gM, gN, gG, and gK, are involved in regulating PRV attachment and fusion [17,18,19,20,21]. Notably, PRV glycoproteins gD, gE, and gI are essential for immune evasion and intercellular transmission of the virus [22]. As shown in Figure 1, PRV gB, gC, and gD are primary targets for inducing neutralizing antibodies [11,23,24], while the other glycoproteins also contribute to stimulating host immune responses [11,25].

Both gE and gB are vital structural components in PRV pathogenesis [23]. gE is critical for viral spread from infected cells to adjacent ones, forming a complex with glycoprotein gI, which enhances virus transmission between cells [26]. This interaction is crucial for PRV’s ability to invade and disseminate within the nervous system, leading to severe neurological symptoms. Additionally, gE may assist in immune evasion by interacting with host immune molecules or modulating cell signaling pathways, thereby helping the virus evade detection and clearance [9,27].

In contrast, PRV gB functions as a class III viral fusion protein that facilitates the fusion of the viral envelope with the host cell membrane, a necessary step for the virus to deliver its genetic material and initiate infection [28,29]. As a major component of the viral envelope, gB is a target of the host immune response. Antibodies against gB can neutralize the virus and prevent infection, although gB may also participate in the virus’s immune evasion strategies [30].

Effective vaccines and diagnostic tools are crucial for controlling PRV and mitigating its impact on animal health. PRV’s gE and gB are central to these efforts [11,31,32]. PRV gB is important for inducing a humoral immune response [33] and is considered a promising candidate for subunit vaccines [30,34]. Meanwhile, gE is a major target in vaccine development, particularly for attenuated vaccines [28]. The gE gene is pivotal for PRV virulence, neuroinvasion, and dissemination within the host’s nervous system. By targeting gE, researchers aim to attenuate the virus and reduce its pathogenicity while still eliciting a protective immune response [9,35]. This paper reviews recent advancements in genetic engineering technologies and their application in developing PRV vaccines and diagnostic tools.

### 1.1. PRV gE

PRV gE is a crucial envelope protein of the Pseudorabies virus (PRV), essential for viral attachment and entry into host cells. The gene that encodes this protein is located in the US (unique short) region of the PRV genome, spanning 1740 base pairs and encoding a protein of 577 amino acids. gE is a type I transmembrane glycoprotein, characterized by a single transmembrane segment that spans the host cell membrane [9,10,36].

Structurally, gE is divided into three domains: the extracellular domain (ETD), the transmembrane domain (TMD), and the cytoplasmic domain (CTD) [26]. Although gE is not essential for viral replication, it is critical for efficient viral transmission between cells (Figure 1) [37]. gE enhances the secondary envelope and facilitates the spread of virions to adjacent cells [9].

PRV gE forms a heterodimer with glycoprotein I (gI), another type I transmembrane protein, through interactions involving its extracellular domains [38,39]. This gE/gI heterodimer is involved in several viral functions, including efficient intercellular propagation in non-neuronal cells and anterograde transmission in synaptically connected neurons [9,37,40,41,42]. Additionally, gE/gI heterodimers can bind to immunoglobulin G (IgG) Fc fragments, influencing the phosphorylation of extracellular signal-regulated kinase 1/2 (ERK1/2) and aiding in immune evasion [9,26,43]. Thus, PRV gE plays a significant role in enhancing viral infectivity and facilitating the virus’s entry into the nervous system.

### 1.2. PRV gB

PRV gB is another key structural component of the PRV envelope. The gene encoding gB is located in the Unique Long (UL) region of the virus’s genome, with a length of 2800 base pairs and encoding a protein of 913 amino acids, including a 58-amino acid cleavable signal peptide [36]. The gB protein exists as a disulfide-bonded complex composed of three subunits: gBa, gBb, and gBc, with gBb and gBc being cleavage products of gBa [44]. Structurally, gB features a trimeric fold, a dimer fusion ring, and an α-helical configuration. gB interacts with cell membranes in a cholesterol-dependent manner via fusion rings [36]. gB is indispensable for viral replication, playing a pivotal role in membrane fusion during viral infection and subsequent transmission [28]. Its function in fusion is critical for the virus’s ability to infect new cells and propagate within the host (Figure 1).

**Figure 1 vaccines-12-01078-f001:**
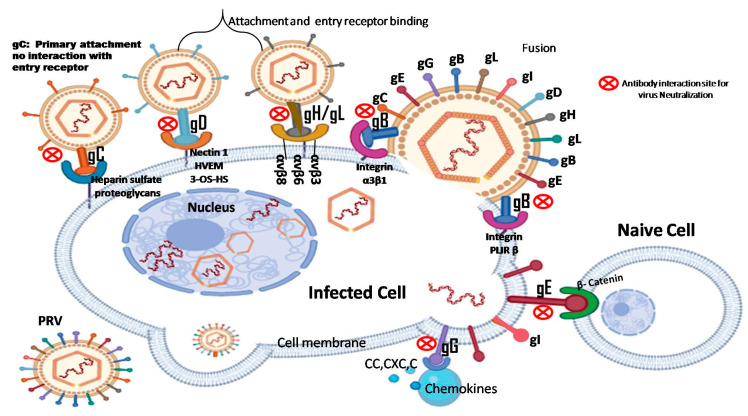
Schematic diagram of interactions between PRV glycoproteins (gB, gC, gD, gE, gG, gH/gL) and various receptors on the host cell membrane. gC: binding to membrane heparin sulfate (HS) proteoglycan as the main attachment receptor; Does not interact directly with the entry receptors on the host cell membrane [45]. gD: Binds to the receptor: nectin-1, HVEM, or 3-OS-HS. gH/gL: transmits activation signals of gD receptor complex to gB; Interaction with integrin promotes endocytosis [46]. gB: membrane fusion; Interaction with PILRβ promotes plasma membrane fusion and induces cytotoxicity of natural killer (NK) cells [47]. gE: Binds to β-catenin of naive cells for intercellular transmission [48]. gG: Binds to chemokine receptors and inhibits chemokine-mediated cell migration [49]. These glycoproteins are targeting that antibody bind to neutralize PRV (Created with Biorender.com).

## 2. Application of PRV Glycoprotein in Vaccine Development

### 2.1. gE as a Vaccine Target

The PRV gE protein is a prominent target in vaccine development due to its significant role in viral pathogenesis. It is highly expressed on the surface of infected cells and integrates into the viral envelope, making it a prime candidate for inclusion in various vaccine types (Table 1), including subunit and nucleic acid vaccines. This high antigenicity and its ability to elicit a strong immune response (Figure 2), make gE a valuable target for vaccine strategies [19,32,49,50,51]

### 2.2. Live Attenuated Vaccines

#### Deletion of gE Gene and Other PRV Proteins from Live Attenuated Vaccines

The gene encoding PRV gE is a priority for developing attenuated vaccines. By deleting or modifying the gE gene in candidate vaccines, researchers can produce weakened (attenuated) viruses for use in live attenuated vaccines [9,29,32,35,52,53,54,55,56,57,58,59,60,61]. A well-known example is the PRV Bartha-K61 vaccine, derived from the Bartha strain. This vaccine was developed through sequential passage [28,62] and features significant deletions in the US region of the genome [29], including the complete removal of gE and US9, and partial deletions of gI and US2 [63,64]. The absence of gE, along with other PRV proteins, contributes to its high safety and immunogenicity [39,65,66,67].

Another critical gene in the context of live attenuated vaccines is PRV thymidine kinase (TK), encoded by UL23. TK is essential for viral replication and pathogenesis [19,68,69]. Deleting the TK gene, often in combination with gE, results in reduced virulence and enhanced immune response [55,70,71].

Experiments with double and triple deletions of gE, gI, and TK have demonstrated promising results [72,73]. For instance, a vaccine with gE/gI double-deletion strains [9,32,55,69,74,75,76,77,78,79,80] induced a high neutralizing antibody titer (1:40.96) and showed non-pathogenicity in suckling piglets. In controlled studies, 90% of vaccinated piglets exhibited complete resistance to PRV, compared to only 20% in the control group [9,65]. Similarly, triple-deletion mutants of gE, gI, and TK have shown even higher protection rates and antibody titers [9,56,81,82,83]. Other vaccine strains like NIA-3, which also involves gE and TK gene deletions [9,78], have shown 95% effectiveness in preventing viral shedding and clinical symptoms [64]. The SA125 strain, with a gE/gI/TK deletion, has demonstrated high efficacy across different ages of piglets, providing 100% survival with only mild symptoms [53]. The HB-2000 and HN1201 strains, both with gE/gI/TK deletions, produced high levels of protective gB-specific antibodies [68,84], offering 100% protection against PRV [83].

### 2.3. Inactivated PRV Vaccine

Inactivated PRV vaccines lacking gE genes are effective in controlling PRV variants. They offer comprehensive protection and can differentiate vaccinated animals from infected ones, making them promising for PRV control [85]. The gE deletion significantly reduces the virus’s ability to invade and spread within the host nervous system, enhancing its susceptibility to immune responses [9,79,86] The PRV/marker gold-KV (SyntroVet) inactivated vaccine has shown high efficacy with 90–95% effectiveness in preventing clinical symptoms and reducing virus shedding [61,85,87,88,89]. The HN1201-ΔgE vaccine, certified for use in China, induces high antibody titers and offers a 95% survival rate [90].

### 2.4. PRV gE Subunit Vaccines

PRV gE subunit vaccines involve the development of vaccines that target the gE of PRV. The goal of this vaccine approach is to induce an immune response in the host using PRV gE, which provides protection against PRV infection. These subunit vaccines are often supplemented with adjuvants to improve immunogenicity [32,84,91].

Several licensed vaccines containing gE, such as D1200/D560 and gD/gI (negative) NIA-3, have been produced and used for swine immunization. These vaccine strains have been shown to be non-toxic to pigs and provide a 100% survival rate [92,93]. The tolvida TK/gpX wild type vaccine strain, a PRV mutant lacking glycoprotein X (gX) and thymidine kinase (TK), induced antibody titers of up to 1:3072, indicating a strong immune response and a 100% survival rate against PRV attack in mice [19,50] Similarly, the PRV-specific genetically engineered S-PRY-155 vaccine (PRV155 Iowa TK/gI/gpX field strain) induced a very high antibody titer of 1:4096, indicating a strong immune response, and provided a 100% survival rate with no virus shedding and no clinical symptoms [61,94].

Omnimarka TK/gIII/gI+ Omnivac (BUK), a modified live vaccine strain from the PRV (Bucharest [BUK]-d13) vaccine strain, was named PRV (delg92/TK). Deletion of the thymidine kinase (TK) and glycoprotein-gIII (g92) genes induced a high antibody titer of 1:3072, providing 100% survival rate, no clinical symptoms, and reduced viral shedding [73,95].

In addition, the Bucharest (BUK) PRV vaccine strain of the TK negative (TK−) deletion mutant Omnivaca TK/gI+, BUK (BUK d13), induced a high antibody titer of 1:2048 and provided a 95% survival rate with mild clinical symptoms and minimal virus shedding [19].

### 2.5. PRV gE DNA Vaccine

PRV gE DNA vaccines are being developed against PRV gE using DNA-based techniques [91]. This approach aims to induce an immune response to PRV by introducing DNA encoding the gE protein into the host cell, leading to the production of gE protein and subsequent immune recognition [39].

Genetically modified DNA vaccines are effective in preventing PRV infection. DNA that codes for gE proteins can stimulate the immune system to produce antibodies and cellular responses that target and neutralize PRV [90]. Studies have shown that genetically modified DNA vaccines can induce a specific immune response against PRV, thereby protecting vaccinated laboratory animals from infection and reducing clinical symptoms [84,96]. However, limited research has been conducted on the effectiveness of these vaccines in the real-world setting.

### 2.6. gB as a Vaccine Target

PRV gB is another important viral protein involved in vaccine development. As an antigen, PRV gB can induce a humoral immune response [11,91] and is considered a promising vaccine candidate [30,34]. PRV gB is a major immunogenetic glycoprotein antigen that induces neutralizing antibodies (Figure 2). Fourteen epitopes have been identified in gBb and gBc, located between aa 59–126, 214–279, and 540–734 in gB (Kaplan, Princeton, NJ, USA) [30,97].

## 3. Development of Immune Response Vaccine against PRV gB

Developing vaccines to elicit immune response against gB is an important area of research, particularly in the context of viral infections [9]. Many herpesviruses [79], including cytomegalovirus (CMV) and herpes simplex virus (HSV), contain the conserved protein gB, which plays a crucial role in viral entry into host cells [98]. PRV gB-based vaccines can be designed to trigger an immune response against this protein, potentially preventing the virus from attaching to and entering the host [11,99]. Recent studies have confirmed that pure PRV gB can induce protective immunity, and several vaccination platforms have introduced gB antigens into the immune system (Table 1).

### 3.1. PRV gB Subunit Vaccines

The developed PRV gB-based subunit vaccine contains only the gB protein or specific epitopes. They are often used in combination with adjuvants to enhance the immune response [1,30].

Anti-PRV gB antibodies recognize epitopes within the gB extracellular domains 59–126, 216–279, and 540–734 residues, focusing on the 540–734 region. To date, 14 epitopes have been identified in these two regions of gB. These epitopes are essential for neutralizing antibody activity, especially in complement-dependent mechanisms [32]. T cell receptors (TCR) recognize immunodominant epitopes presented by MHC molecules, leading to a strong cell-mediated immune response. DNA vaccines encoding gB effectively induce these responses, highlighting the importance of gB in T cell activation [100].

Studies have shown that immunization with a gB-based subunit vaccine induces protective immunity [84]. For example, a gB subunit vaccine trial achieved 100% survival in immunized pigs vaccinated with the PRV HNLH strain [24]. When administered nasally, the PRV gB vaccine induces the production of IgA and IgG antibodies, thereby protecting animals from nasal exposure to lethal doses of virulent PRV [101,102].

Studies have also shown that subunit vaccines containing gB, in combination with adjuvants such as granulocyte-macrophage colony-stimulating factor (GM-CSF), can trigger high levels of neutralizing antibodies (titer 1:4096) and provide 100% protection against PRV attacking the variants [84]. In addition, the gB subunit vaccine prepared from PRV strain SD-2017 had a neutralizing antibody titer of 1:2048 and a survival rate of 98% [84]. The gB subunit vaccine of the HN1201 strain induces a high antibody titer of 1:4096 and provides a 100% survival rate against PRV [1].

The gB subunit vaccine, which has high antibody levels in vaccinated animals, has shown promising efficacy in providing immunity against PRV infection and may aid virus control and eradication [51,84,89,103].

### 3.2. PRV gB DNA Vaccine

The PRV gB-based DNA vaccine approach uses the DNA encoding PRV gB to induce an immune response against PRV infection. This vaccine strategy was designed to stimulate the production of gB proteins within host cells, thereby triggering an immune response that prevents PRV infection [60,64,89,104].

Studies have shown that gB-based DNA vaccines are effective in preventing PRV infections. Immunizing pigs with purified gB-based DNA vaccines produces a high antibody titer (1:3072) and provides complete protection against lethal PRV attacks in mice and pigs [105]. Pigs inoculated with plasmid DNA encoding the gB key immunogenic antigen produce a strong cell-mediated immune response [9,106,107]. Piglets, in particular, can achieve strong cell-mediated immunity, including cytotoxic T cell responses, by inoculation with plasmid DNA encoding gB. Booster vaccination enhances immune response. In addition, gB-DNA vaccines can effectively inhibit the release of the virus shortly after a challenge infection [106,108,109,110].

### 3.3. Viral Vector Vaccines

Viral vector vaccines use vectors to deliver gB antigens into cells, leading to the expression of gB proteins and an immune response against PRV [31]. Studies have shown that viral vector vaccines, including those using recombinant baculoviruses [111,112,113,114], bovine herpes virus 1 (BHV-1) [115], poxviruses [2,116,117], and canine herpesvirus (CHV) [118] can produce high antibody titers and offer protection against PRV [1,61,109,111,113,119,120].

In conclusion, the development of PRV vaccines utilizing glycoproteins like gE and gB has demonstrated significant progress and potential. Various vaccine strategies, including live attenuated, inactivated, subunit, DNA, and viral vector vaccines, offer different advantages in terms of safety, efficacy, and protection against PRV. Continued research and development are essential for optimizing these vaccines and ensuring effective control of PRV outbreaks.

**Figure 2 vaccines-12-01078-f002:**
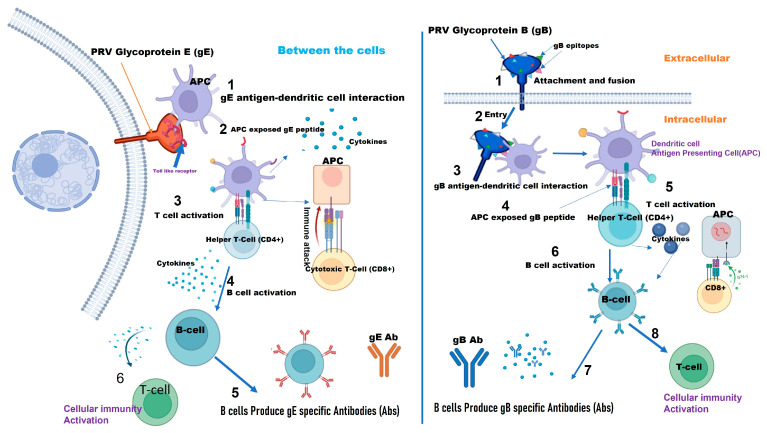
Diagram of pathways depicting how PRV (Pseudorabies Virus) glycoprotein B (gB) and PRV glycoprotein E (gE) induce an immune response, highlighting viral entry, immune cell activation, and antibody production (Created with BioRender.com).

## 4. Comparative Analysis of Vaccine Effectiveness, Safety, and Field Application

PRV vaccines vary significantly in terms of effectiveness, safety, and field applicability, depending on their design and formulation.

Live Attenuated Vaccines: The Bartha-K61 strain of PRV is a prominent example of a live attenuated vaccine, known for its high efficacy. It induces strong humoral and cellular immunity, significantly reducing viral shedding and preventing clinical symptoms [32,79]. However, these vaccines pose certain risks, such as the potential for reversion to virulence, particularly in immunocompromised animals, and may be less effective against emerging PRV variants [28,64,68,121].

Inactivated Vaccines: Inactivated vaccines, such as the PRV/marker gold-KV, offer a safer alternative by utilizing non-infectious viral particles and incorporating gE gene deletion to reduce neuro-invasiveness [68]. While these vaccines generally require multiple doses to achieve comparable levels of immunity and are dependent on reliable cold-chain logistics for large-scale application, they are regarded as safer due to the absence of live virus [68,85,89]

Subunit Vaccines: Subunit vaccines, which target specific viral proteins like gB, are praised for their safety profile and suitability for vulnerable animals due to their minimal risk of adverse reactions. Despite their variable immunogenicity and reliance on adjuvants to enhance efficacy, subunit vaccines are effective for targeted vaccination campaigns [84,91] DNA Vaccines: DNA vaccines, though still experimental, show promise for inducing specific immune responses without the risk of live virus introduction. These vaccines have potential, but further research is necessary to evaluate their field efficacy and long-term safety [9,64,91,122].

In conclusion, the choice of PRV vaccine should be tailored to the specific PRV strain, the health status of the animals, and logistical considerations. While live attenuated vaccines offer high efficacy, inactivated and subunit vaccines provide safer alternatives. Ongoing research and development are crucial to optimizing vaccine strategies and enhancing PRV control.

**Table 1 vaccines-12-01078-t001:** List of different types of gE and gB-based PRV vaccine approaches and their characteristics.

Vaccine Approaches	Characteristics	Protective Efficacy	Effective on	Vaccine Status
gE Live attenuated vaccine [50,57,123]	Attenuated strain with gE gene deleted	High	Classical and variant strains	Approved
gE Inactivated PRV vaccine [85,90,124,125]	Inactivated strain with gE gene deleted	Moderate	Classical strains	In development
gE subunit vaccine [61,88,91,126]	Recombinant gE protein Generally safe	Moderate	Classical strains	Approved
gE DNA vaccine [76,89]	DNA encoding gE antigen	Low	Classical strains	In development
gB subunit vaccine [127,128,129]	Recombinant gB protein Generally safe	High	Classical and variant strains	Approved
gB DNA vaccine [62,129]	DNA encoding gB antigen	Low	Classical strains	In development
gB live attenuated vaccine [59,130]	Attenuated strain with gB gene deleted	High	Classical and variant strains	Approved
Recombined PRV gB vaccine [99,131]	Recombinant virus expressing the gB antigen, Expressing both native and foreign antigens	High	Classical and variant strains	Approved

## 5. PRV Glycoproteins in Diagnosis Kit Development

PRV glycoproteins E and B are used in diagnostic kit development (Table 2) to detect PRV infections [130,131]. These antigens can be used to detect antibodies produced against PRV gE and gB using serological diagnostic tests, such as enzyme-linked immunosorbent assays (ELISAs) or neutralization tests. Differential diagnostic tests [132] allow veterinarians to distinguish PRV-infected pigs from those vaccinated with labeled vaccines through serological analysis. The availability of a missing marker vaccine with an appropriate differential diagnostic test [19,85] has encouraged initiatives to manage and eradicate PRV in most parts of the world [123].

### 5.1. ELISA Kit Development

Many ELISA kits have been developed using gE proteins, such as the direct gE ELISA kit [133,134,135], which can distinguish antibodies produced in response to gE deletion vaccines or wild viral infections. These laboratory-based methods have excellent sensitivity (>99.4%) and specificity (>98.4%) [64,119,126,136,137,138]. Indirect PRV gE ELISA kits, such as the PrioCHECK™ PRV gE Antibody ELISA Kit and SVANOVIR PRV gE-Ab, are qualified diagnostic tests used during eradication procedures and import and export of certified pigs. This indirect PRV gE-ELISA had 99.1% sensitivity and 99.6% specificity [61,129] At the same time, and a gE CTD-specific blocking ELISA kit was developed to distinguish between cows infected with wild BHV-1 strains and those infected with gE CTD-deficient strains. The gE CTD protein can act as either a coating antigen or an indicator antibody with high sensitivity (>99.0%) and specificity (>99.9%) [9,57,139]. In addition, an indirect sandwich ELISA kit based on ygEN31-270 has been used to detect gE specific antibodies in pig serum samples with high sensitivity (>89.53%) and specificity (>90.32%) [119,137,140].

In contrast, researchers have developed an ELISA diagnostic kit using PRV gB protein, which can identify specific antibodies to gB in pig serum or plasma, indicating exposure to PRV (Table 2). BacgB ELISA is a valuable tool for distinguishing between vaccinated piglets and those infected with strains other than those used in vaccines. BacgB ELISA showed exceptional sensitivity (97.3%) and specificity (98.5%) for detecting PRV gB antibodies [98,141,142].

Overall, the PRV gE/gB ELISA kit is a valuable tool for veterinarians, diagnostic laboratories, and researchers to screen for PRV exposure in pig herds. By detecting gE/gB protein antibodies, this kit contributes to the monitoring, diagnosis, and control of PRV infections in pigs [143,144].

### 5.2. Neutralization-Assay

Neutralization tests based on PRV gE/gB evaluate neutralizing antibodies against PRV strains [84] Neutralizing antibodies targeting PRV gB mainly target epitopes in the 540–734 region of PRV gB residues. These antibodies target epitopes within this region, including the upper part of the central helix of domain III and the entire gB domain IV/truncated version [11]. Neutralizing antibodies targeting PRV gB residues 540–734 can block PRV entry in the presence of a complement, highlighting the importance of this region in the anti-PRV immune response [145]. Meanwhile, PRV gE neutralizing antibody/monoclonal antibody (mAb) 1H5 recognizes epitopes within PRV gE protein residues 67–72 (67RRAG70). This region is conserved in 244 different strains of PRV, indicating its potential as a target for broad-spectrum PRV diagnosis and treatment [146]. The identification of linear epitopes recognized by mAb 1H5 provides insights into the antigenic determinants of PRV gE. In this study, mAb 1H5 was prepared using a new immunization and screening strategy, and the antibody responded well to PRV strains in an immunofluorescence assay (IFA) and indirect ELISA, demonstrating its usefulness in detecting PRV infections [143,146].

Tests based on PRV glycoproteins are often used to distinguish PRV antibodies in infected and vaccinated animals. This approach has played an important role in pathogen eradication programs around the world [29,61,67,121].

The immunodominant region of gE expressed in Pichia Pastoris can induce monoclonal antibodies, called recombinant gE antibodies [119]. This monoclonal antibody specifically recognizes classical PRV and mutant strains [146]. In contrast, an AGAR gel immunodiffusion (AGID) test (Table 2) for the detection of PRV antibodies was established using PRV gB, and the derived AGID sensitivity and specificity were 95% and 96.6%, respectively [61]. Therefore, PRV gB has become a valuable source of antigen for the detection of PRV antibodies in AGID tests, particularly when using the Bartha strain [61,147].

The PRV gB IndianaS strain [89] and gB Tolvid strain [19] are commonly used in viral neutralization (VN) tests to assess the ability of antibodies present in serum samples to neutralize PRV. These strains were used for serological analysis to determine the levels of neutralizing antibodies in test serum, thereby providing valuable insights into the immune response to PRV infection or vaccination [50,148].

Researchers and veterinarians can conduct neutralization tests to evaluate the protective efficacy of the vaccine against the PRV strain and neutralizing ability of the antibody. This could help advance vaccine development, monitor its efficacy, and further understand the immune response to PRV infection [84].

### 5.3. Antibody Test Kit

The PRV Antibody Test Kit is an immunoassay kit that detects gB and gE antibodies against PRV in pig serum. The BioCheck’s antibody test kit can distinguish between infected and vaccinated animals and is a valuable tool for herd management and eradication applications [149].

The PRV gB Antibody Test Kit has a high correlation with serum neutralization (SN) tests, providing reliable results for PRV gB antibody detection. This provides a practical and effective method for monitoring PRV infection in pigs [150]. The PRV gE Antibody Test Kit detects PRV gE antibodies and identifies animals that have been vaccinated against PRV or are naturally infected with the virus. The performance of the PRV gE Antibody Test Kit is essential for distinguishing between vaccinated and wild-type PRV-infected animals, helping to control and eradicate PRV [51].

The latex agglutination test (gE-LAT) (Table 2), which is based on PRV gE, is classified as an immunoassay to detect antibodies specific to the PRV gE antigen. gE-LAT uses latex particles coated with the gE antigen from PRV [140] to distinguish between PRV-infected pig sera and pigs vaccinated with the gE deletion vaccine. The test showed high diagnostic specificity (96.77%) and sensitivity (95.76%) [140]. It can be used as a routine screening method for the differential diagnosis of PRV infections [125].

The PRV gE Antibody Test Kit provides a more practical and effective method for detecting gE-specific antibodies than other diagnostic methods such as neutralization tests (Table 2). It provides a reliable and rapid method for screening large numbers of samples, making it suitable for routine monitoring of pig herds [149].

Veterinarians, farmers, and individuals involved in raising pigs use PRV gE/gB antibody test kits to screen for PRV infection, confirm that the herds are disease free, and monitor control plans. The test is suitable for anyone seeking fast and accurate results within 10–30 min. The sensitivity and specificity of the PRV antibodies in pig serum were >98.1% and >98.1%, respectively. In addition, it is cost-effective, making it a valuable tool for managing and controlling PRV infections [150,151,152]. While the PRV Antibody Test Kit is effective in detecting PRV infections and managing outbreaks, it currently cannot detect new variant strains of PRV. Thus, targeting gE and gB glycoproteins in the development of antibody detection kits for these variant strains is crucial to improve their diagnostic capabilities.

### 5.4. PCR Detection Kit

The development of PRV gB and/or gE based polymerase chain reaction (PCR) kits involves utilizing the gB and/or gE genes of PRV as targets for PCR assay. This kit was designed to provide a sensitive and specific method for detecting PRV infection by detecting the gB/gE gene of the virus. By designing specific primers and probes that bind to the gB and gE gene regions [153,154,155], PCR assay kits can amplify and detect PRV DNA in a sample by designing specific primers and probes that bind to gB and gE gene regions. These probes and primers can accurately detect and identify wild-type and gE defective vaccine strains [156].

PRV gE PCR detection kit uses multiple real-time polymerase chain reaction (EGRT-PCR) (Table 2) to distinguish wild-type and gE-deficient strains based on PRV gE specific temperature (Tm) peaks [9,157,158,159,160]. PCR assays use gE DNA to distinguish between vaccinated and wild-type PRV-infected animals [9,61]. In addition, the PRV gB PCR detection kit uses real-time fluorescent PCR technology to detect PRV (gB gene) RNA in tissue samples such as tonsils, lymph nodes, and spleen, and liquid samples such as vaccines and pig blood [144].

In the PCR detection kit, gB and gE genes were used as targets to reliably identify PRV, which contributed to effective monitoring, diagnosis, and control of PRV in pigs [151,161].

### 5.5. Rapid Test Kit

The development of a rapid PRV assay kit based on PRV gB and gE involves the use of these two glycoproteins as targets for detecting PRV infections in pig herds [85,162]. The test kit was designed to provide sensitivity (98.1%) and specificity (95.4%) for diagnosing PRV infection, enabling timely implementation of treatment and control measures. This test kit typically uses side-flow immunochromatography, which can produce rapid results in a short time (usually 5–10 min) (Table 2). Veterinarians, farmers, and other individuals involved in pig production can use the test to quickly identify PRV infection, take appropriate action to prevent the transmission of the virus, and minimize its impact on pig health and production [163].

**Table 2 vaccines-12-01078-t002:** Comparison of PRV gB and gE based diagnostic Kits.

Diagnostic Kit	Characteristics	Performance
PRV gE Antibody Test Kit [150,164]	Detect PRV gE antibody in serum Solid-phase microchip platform	High Sensitivity (98.1%) and specificity (98.8%), Suitable for the laboratory and field test, 10–30 min test duration, Low cost
PRV gE Real-time PCR Kit [163,165,166]	Detect PRV gE gene sequence in swine serum, plasma, tissue, sperm, and environmental samples. Used for DIVA strategy	Very high sensitivity (100%) and specificity (100%). Suitable for laboratory test 1–4 h test duration, Moderate to high cost
PRV gB Antibody Test Kit (BioChek) [126]	Semi-quantitative ELISA Detect PRV gB antibody Confirms herd disease.	High Sensitivity (98.1%) and specificity (98.1%) Suitable for field and laboratory test, 10–30 min test duration, Low cost
PRV gB ELISA [143,144]	Detect PRV gB antibody in serum, Provides quantitative measurement of antibodies	High Sensitivity (>80.9%) and specificity (>96.4%) Suitable for laboratory test 2–4 h test duration, Moderate to high cost
New immuno chromatographic strip [167]	Rapidly detect PRV gB antibody, Easily applicable with no special skill	High Specificity (95.4%) and sensitivity (98.1%) Suitable for field test 5–10 min test duration, Low cost
PRV gB real time PCR [126,163,166,168]	Detect PRV gB gene in tissue, blood, and vaccine samples. Aids disease surveillance and control.	Very high sensitivity (100%) and specificity (100%), Suitable for laboratory test, 1–4 h test duration, Moderate to high cost
PRV gE-based Latex Agglutination Kit (gE-LAT) [169,170]	Detect PRV gE antibodies Provides rapid results Reliable and easy to use	High sensitivity (95.76%) and high specificity (96.77%) Suitable for field test, 30–90 min test duration, Moderate cost
PRV gB Agar gel immuno diffusion (AGID) test kit [147,171]	Detect PRV gB antibodies in serum. Simple, cost-effective, and reliable	High sensitivity (95%) and specificity (96.6%) Suitable for laboratory test 24 to 48 h test duration, Moderate cost

## 6. Challenges and Future Perspectives

Recently, the virulence of PRV and failure of vaccine protection have been enhanced due to genetic mutations and recombination, resulting in the emergence of more pathogenic variants. Recent studies confirmed that PRV genomic recombination is essential for PRV to evade immune recognition, resulting in antigenic mutations or the emergence of new strains with increased virulence. One study identified a new recombinant virulent PRV strain, HN-2019, derived from classic PRV (e.g., Ea) and HB-98 vaccine strains [172].Virulence of the recombinant strain in mice was similar to that of the parent strain. This suggests that large-scale use of live attenuated vaccines may increase the likelihood of recombination, potentially leading to the emergence of novel PRV strains with different virulence levels and leads for the vaccine failures [1,35,173,174,175,176]. Bartha-K61, the first attenuated PRV strain with multiple protein deletions, has been widely used as a live attenuated vaccine world wide to eradicate PRV in domestic pigs. However, in October 2011, a severe PRV outbreak occurred in pigs vaccinated with the genetically deficient Bartha-K61 classic vaccine strain in northern China [3]. The inability of Bartha K61 to provide absolute protection against PRV-variant infections remains a concern [64]. Despite the efforts of scientists to develop diagnostic methods and vaccines in recent years, PRV remains a serious infectious disease, especially in Asian countries such as China, which has had a significant impact on the pig industry. The potential harm caused by PRV to human health has once again attracted international attention [164].

To overcome this obstacle, further research is required to prevent and control the disease. Several strategies can be considered, including the development of multivalent vaccines that combine multiple PRV antigens, such as glycoproteins gE and gB, to provide broader protection. Gene-deleted vaccines, which remove virulence genes, such as gE, gI, and TK, reduce the risk of reversion to virulence. The use of novel adjuvants can enhance subunit vaccine effectiveness, whereas continuous monitoring of circulating strains allows for timely adaptation of vaccines. DNA and RNA based vaccines offer flexible platforms that can be updated rapidly to address new variants. Focusing on conserved regions of PRV glycoproteins for broad-spectrum vaccine development and employing a prime-boost strategy with live attenuated and inactivated vaccines can further mitigate vaccine resistance. These studies may reveal additional functions of PRV glycoproteins, enhance our understanding of PRV pathogenic molecules, and validate the efficacy of alternative glycoproteins as potential vaccines and diagnostic agents.

## 7. Conclusions

PRV remains a severe and economically damaging disease affecting pigs, with significant implications for the pig industry globally. PRV glycoproteins are central to the development of vaccines and diagnostic tools due to their role in eliciting both cellular and humoral immune responses. Among these glycoproteins, PRV gB and gE are particularly important. PRV gB is vital for inducing protective immunity and is a key component in vaccine development due to its strong antigenicity and surface presence. Conversely, PRV gE is used for differential diagnostics and as a target antigen in various diagnostic aids. The development of attenuated PRV strains with reduced virulence, such as the Bartha strain, represents a significant advancement in vaccinology. Recombinant approaches have also proven effective in producing PRV gB in vitro, and the integration of gE and gB into other organisms’ genomes holds promise for generating protective immunogens.

To address emerging PRV variants and enhance vaccine and diagnostic efficacy, ongoing research into PRV glycoproteins and their recombinant applications is essential. This will help develop effective vaccines and diagnostic tools capable of providing comprehensive protection against evolving PRV strains.

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
