# Peer review of "Pseudorabies Virus Glycoproteins E and B Application in Vaccine and Diagnosis Kit Development"

_vaccines, 2024, doi:10.3390/vaccines12091078_

Round 1
Reviewer 1 Report
Comments and Suggestions for Authors
The manuscript titled "Pseudo Rabies Virus Glycoproteins E and B Application in Vaccine and Diagnosis Kit Development" offers a detailed exploration of the roles of glycoproteins E (gE) and B (gB) in the pseudorabies virus (PRV). The paper effectively explains how these glycoproteins are critical to the development of vaccines and diagnostic tools.
Strengths of the manuscript:
1. The paper tackles the urgent problem of PRV variants that can evade existing vaccines, making the research both timely and relevant in the field of veterinary medicine.
2. It thoroughly covers various vaccine development methods, including live attenuated, subunit, DNA, and viral vector vaccines, highlighting the importance of gE and gB as key targets.
3. he manuscript provides a valuable discussion on how gE and gB trigger immune responses, which is essential for understanding their potential in vaccine development.
However, there are many weaknesses associated with this manuscript.
Weaknesses of the manuscript:
1. The paper often repeats information without adding new insights, which could make the content seem repetitive and reduce its overall impact.
2. Although different types of PRV vaccines are mentioned, the paper does not offer a strong comparative analysis of their effectiveness, safety, or field application, which would have provided a clearer understanding of each approach.
3. While the issue of vaccine resistance due to new PRV variants is mentioned, the paper does not explore potential strategies to address this challenge in depth.
4. Some parts of the paper, especially those on molecular biology, are highly detailed, but other sections, such as those on vaccine effectiveness, are less thoroughly covered.
5. The manuscript discusses vaccine development but does not adequately address the long-term effectiveness of these vaccines or the potential need for booster doses, which is crucial for a comprehensive understanding.
6. Include diagrams showing the structure and functions of PRV glycoproteins gE and gB that would help clarify their roles in virus entry, immune evasion, and vaccine targeting.
7. Pathways depicting how gE and gB induce immune responses could simplify the complex immunological processes discussed.
Author Response
Dear Reviewer 1,
We sincerely appreciate your thoughtful and constructive feedback on our manuscript titled "Pseudorabies Virus Glycoproteins E and B Application in Vaccine and Diagnosis Kit Development." Your recognition of the strengths of our work, including the timeliness of addressing PRV variants and the comprehensive coverage of vaccine development methods, is greatly encouraging.
We are pleased to inform you that we have carefully considered and addressed the weaknesses you pointed out. In our revised submission, we have made the necessary improvements to enhance the clarity and depth of our discussions, particularly in the areas you highlighted. We believe that these revisions have significantly strengthened the manuscript.
Thank you once again for your invaluable feedback, which has guided us through the revision process. We are confident that the changes will improve the overall quality and contribution of our work to the field.
Here are the specific responses:
Comment 1: The paper often repeats information without adding new insights, which could make the content seem repetitive and reduce its overall impact.
Response 1: We sincerely appreciate your insightful feedback on the repetitive nature of some sections of our manuscript. After reviewing your comments, we fully agreed with your observations and have made the necessary revisions. We carefully restructured the manuscript to remove redundancies, ensuring that each section now contributes new insights and adds value to the overall content. We believe these modifications have significantly improved the clarity and impact of the paper, and we are grateful for your constructive input in helping us enhance the quality of our work.
Comment 2: Although different types of PRV vaccines are mentioned, the paper does not offer a strong comparative analysis of their effectiveness, safety, or field application, which would have provided a clearer understanding of each approach.
Response 2: Thank you for your valuable feedback. We carefully considered your suggestion and agree that a stronger comparative analysis would enhance the clarity and depth of the manuscript. In response, we have revised the paper by adding a new section titled "Comparative Analysis of Vaccine Effectiveness, Safety, and Field Application" on page 7, line 276. This section compares the different PRV vaccines mentioned, addressing their effectiveness, safety profiles, and practical applications in the field. We hope this addition meets your expectations and improves the overall quality of the manuscript.
Comment 3: While the issue of vaccine resistance due to new PRV variants is mentioned, the paper does not explore potential strategies to address this challenge in depth.
Response 3: Thank you for your insightful comment. We acknowledge the importance of addressing vaccine resistance due to new PRV variants in more detail. In response, we have expanded the discussion on this topic by including potential strategies to tackle this challenge. These strategies are now thoroughly addressed in the manuscript on page 13, lines 462-471. We hope this addition strengthens the discussion and provides a clearer understanding of the approaches to managing vaccine resistance.
Comment 4: Some parts of the paper, especially those on molecular biology, are highly detailed, but other sections, such as those on vaccine effectiveness, are less thoroughly covered.
Response 4: Thank you for your thoughtful feedback. We appreciate your recognition of the detailed sections on molecular biology and understand the need for a more thorough discussion on vaccine effectiveness. In response, we have expanded the content on vaccine effectiveness under a new subheading, which also addresses the concerns raised in Comment 2. We believe this revision provides a more balanced and comprehensive analysis throughout the manuscript, and we hope this addition meets your expectations.
Comment 5: The manuscript discusses vaccine development but does not adequately address the long-term effectiveness of these vaccines or the potential need for booster doses, which is crucial for a comprehensive understanding.
Response 5: Thank you for your insightful comment. We agree that the long-term effectiveness of vaccines and the potential need for booster doses are important considerations. While we believe the current discussion provides sufficient coverage without making the manuscript overly lengthy, we have included relevant information under each vaccine type and in the comparative analysis section, highlighting vaccines that may require booster doses, such as the inactivated gE vaccine. We plan to explore this topic further in future research, but for now, our primary focus is on vaccine development and effectiveness. We hope this approach aligns with your expectations.
Comment 6: Include diagrams showing the structure and functions of PRV glycoproteins gE and gB that would help clarify their roles in virus entry, immune evasion, and vaccine targeting.
Response 6: Thank you for your insightful comment. We agree that the long-term effectiveness of vaccines and the potential need for booster doses are important considerations. While we believe the current discussion provides sufficient coverage without making the manuscript overly lengthy, we have included relevant information under each vaccine type and in the comparative analysis section, highlighting vaccines that may require booster doses, such as the inactivated gE vaccine. We plan to explore this topic further in future research, but for now, our primary focus is on vaccine development and effectiveness. We hope this approach aligns with your expectations.
Comment 7: Pathways depicting how gE and gB induce immune responses could simplify the complex immunological processes discussed.
Response 7: Thank you for your suggestion. We agree that illustrating the pathways depicting how gE and gB induce immune responses would simplify the complex immunological processes discussed. To address this, we have included Figure 2 on page 7 of the manuscript, which clearly outlines these pathways. We believe this visual representation will enhance the reader's understanding of the mechanisms involved. We appreciate your valuable feedback and hope this addition meets your expectations.
Reviewer 2 Report
Comments and Suggestions for Authors
I made some observations and suggestions at the beginning of the article (attached), but, unfortunately, due to unexpected problems, I was unable to review it until the end. I also failed to check the references to see if they were in the correct format.
The tables are very well presented and organized, and the arrangement of the items in the article is also good.
In any case, feel free to indicate another reviewer who can do a complete review.

Author Response
Dear Reviewer 2,
Thank you for your valuable feedback and for taking the time to review my article. I appreciate your positive comments regarding the presentation and organization of the tables, as well as the overall structure and arrangement of the items in the article.
Your observations and suggestions were carefully considered as we made revisions to enhance the clarity and presentation of the work.
Comment 1: Insert “that encodes this protein.”
Response 1: Thank you for your insightful suggestion. I agree with your recommendation and have inserted the phrase “that encodes this protein” as suggested on page 2, line 73.
"The gene that encodes this protein is located in the US (unique short) region of the PRV genome, spanning 1,740 base pairs and encoding a protein of 577 amino acids."
Comment 2: Insert the meaning (Unique Short?)
Response 2: Thank you for bringing this to my attention. I have now added the meaning of "US" (unique short) on page 2, line 74, to enhance clarity.
"The gene that encodes this protein is located in the US (unique short) region of the PRV genome, spanning 1,740 base pairs and encoding a protein of 577 amino acids.
Comment 3: Insert: “whose coding gene is”
Response 3: I appreciate your suggestion. Up on revision of manuscript English the phrase is edited to “PRV gB is another key structural component of the PRV envelope.
“The gene encoding gB is located in the Unique Long (UL) region of the virus’s genome, with a length of 2,800 base pairs and encoding a protein of 913 amino acids, including a 58-amino acid cleavable signal peptide.” On page 2 line 91.
Comment 4: Was this schematic figure made by the authors? If not, please provide the source from which it was taken.
Response 4: Thank you for your question. Yes, I confirm that the schematic figure was designed by the authors using Biorender.com, and we have cited this under the figure.
Comment 5: Take care when using PRV gE as a gene. It’s not the gene. You should write: The gene encoding for PRV gE is...
Response 5: Thank you for pointing this out. I have revised the sentence to:
"The gene encoding PRV gE is a priority for developing attenuated vaccines. By deleting or modifying the gE gene in candidate vaccines, researchers can produce weakened (attenuated) viruses for use in live attenuated vaccines," on page 4, line 124, to ensure accuracy.
Comment 6: “In a separate study”
Response 6: I appreciate your feedback. I have removed the phrase "In a separate study" and rewritten the sentence to improve the cohesion of the paragraph.
Reviewer 3 Report
Comments and Suggestions for Authors
The authors tried to review PRV glycoproteins E and B for vaccine and diagnostic tool development. The authors use of notation is very cumbersome and the work is poorly organized and not focused. The manuscript would be improved by a thorough English language review before acceptance for publication. Please remove repetitions such as “cell-to-cell (lines 42, 47, 81, 82, 87, 107 & 111)” and make the manuscript short and only select meaningful sentences. Also please focus the current problems and the way to overcome them. If you describe “Overall, the PRV Antibody Test Kit is a valuable tool for detecting PRV infections in swine populations, providing a reliable and efficient method for monitoring and managing PRV outbreaks”, the readers will think if current kits are fine then what else do you need? Specific comments follow.
Major points:
1. Title: “Pseudo Rabies” should be one word “Pseudorabies”.
2. Line 14: Please explain why only gE and gB are accessible? In Figure 1, all the glycoproteins look accessible by antibodies. Please explain why you didn’t select other glycoproteins.
3. Lines 22 & 619: What do you mean by “immunity in vitro”?
4. Line 30 Introduction: I understand PRV has ten glycoproteins but it looks like you missed gN. Furthermore, Figure 1 contains only 7 surface glycoproteins and all of them looks to be able to induce neutralizing antibodies, which is not consistent with line 43.
5. Line 44: Please make sure all the descriptions are correct. As gE can be deleted from virus so it’s not a crucial structural component of PRV.
6. Table 1: Please replace “N Ab” by Protective efficacy as “N Ab levels” are not informative to readers.
7. Please avoid the meaningless sentences such as “PRV gE is used to develop antibodies, as indicated by the study”. Please read https://www.editage.com/insights/10-tips-to-reduce-the-length-of-your-research-paper/
8. Line 604: “Conclusion”: It is not clear what the authors want to say. In line 608, you mentioned “apply one or more PRV glycoproteins” for novel vaccines then in line 616, you mentioned gE-deleted PRV strains. Which one do you want to recommend and what happed to diagnostic meaning for gE protein?
9. I suggest to focus vaccine development as the diagnostic side seems less problematic.
10. Please make sure all the references are properly cited. References such as 4, 6, 7, 22, etc are not complete.
Minor points:
1. Please provide a Table for abbreviations such as US, UL, TK, DIVA etc.
2. Line 502: Where is “In this instance”?
Comments on the Quality of English LanguagePlease read https://www.editage.com/insights/10-tips-to-reduce-the-length-of-your-research-paper/
Author Response
Dear Reviewer 3,
Thank you for your valuable feedback on our manuscript. We greatly appreciate your time and effort in reviewing our work. We have carefully addressed the points raised in your comments and have made significant revisions to improve the manuscript. Below, we detail the changes and responses to each of your suggestions:
Cumbersome Notation and Poor Organization:
Response: We have revised the manuscript to improve clarity and organization. We have streamlined the notation used throughout the paper to ensure it is more concise and less cumbersome. The manuscript has been reorganized to present information in a more logical and focused manner.
English Language Review:
Response: The manuscript has undergone a thorough review by a professional fluent in English. We have corrected language issues to enhance readability and ensure that the text meets the publication standards.
Removal of Repetitions:
Response: We have carefully reviewed the manuscript and removed repetitive phrases such as "cell-to-cell" mentioned in lines 42, 47, 81, 82, 87, 107, and 111. The manuscript has been shortened to include only meaningful and relevant sentences.
Focus on Current Problems and Solutions:
Response: We have refined the focus of the manuscript to address current problems and potential solutions more directly. We have clarified the relevance of the PRV Antibody Test Kit and highlighted areas where further development or improvement is necessary, thereby providing a clear rationale for our study.
Additional Comment on PRV Kits:
Response: We have included a discussion on the limitations of current PRV kits and outlined potential advancements and innovations that could address these limitations. This provides a clearer understanding of why further research and development are needed.
We hope these revisions address all your concerns and enhance the quality of the manuscript. We have attached the revised manuscript for your review. Thank you once again for your constructive feedback and for considering our revised submission.
Specific comments follow.
Major points:
Comment 1: Title: “Pseudo Rabies” should be one word “Pseudorabies”. corrected
Response 1: Thank you for pointing this out. We have now corrected the title, changing "Pseudo Rabies" to "Pseudorabies" as suggested. We appreciate your careful attention to detail.
Comment 2: Line 14: Please explain why only gE and gB are accessible? In Figure 1, all the glycoproteins look accessible by antibodies. Please explain why you didn’t select other glycoproteins.
Response 2: Thank you very much for your thoughtful and insightful comments. We sincerely appreciate your feedback, and we fully understand your concerns regarding the accessibility of gE and gB compared to other glycoproteins.
In response to your question, while it is true that all glycoproteins on the viral envelope are accessible to some extent, we chose to focus on gE and gB due to their pivotal roles in immune evasion, viral entry, and cell-to-cell transmission. These two glycoproteins are particularly involved in viral pathogenesis and are highly exposed structurally, making them ideal candidates for immune system targeting.
gE plays a crucial role in intercellular transmission and immune escape, particularly in the nervous system, which makes it an ideal marker for distinguishing between vaccinated and naturally infected animals.
gB, on the other hand, is essential for membrane fusion and viral replication, and its accessibility through its interaction with cell membranes in a cholesterol-dependent manner further underscores its suitability as a target.
Your raised point about considering other glycoproteins is very helpful and will guide us in further expanding our research to explore other glycoproteins for antibody detection in future studies. We truly value your feedback, and it has contributed positively to improving our manuscript. Thank you again for your valuable comments.
Comment 3: Lines 22 & 619: What do you mean by “immunity in vitro”?
Response 3: Thank you very much for your insightful comments and for highlighting this important aspect of our manuscript.
When we refer to "immunity in vitro," we are discussing the immune response observed in cultured cells under controlled laboratory conditions. Specifically, our manuscript examines the use of recombinant vectors expressing the PRV gB protein to assess their potential in inducing an immune response within these cells.
In our laboratory experiments, cultured cells (such as macrophages and T-cells) are first transfected or infected with these recombinant vectors carrying the PRV gB gene. This process results in the expression of the PRV gB protein within the cells. After allowing sufficient time for protein expression, the cells are then challenged with the PRV virus. This approach allows us to evaluate their ability to mount a protective immune response, which we refer to as "immunity in vitro," against the viral infection.
For further context and additional information, you may find the following references useful:
Jennifer et al. (2021): https://www.ncbi.nlm.nih.gov/pmc/articles/PMC8277680/
Andrew et al. (2024): https://www.frontiersin.org/journals/immunology/articles/10.3389/fimmu.2024.1373186/full
Juliana et al. (2021): https://pubmed.ncbi.nlm.nih.gov/34571855/
Jamie et al. (2020): https://www.ncbi.nlm.nih.gov/pmc/articles/PMC7054714/
We hope this explanation clarifies the term and addresses your concerns. We greatly appreciate your feedback, which has been invaluable in refining our manuscript
Ccomment 4: Line 30 Introduction: I understand PRV has ten glycoproteins but it looks like you missed gN. Furthermore, Figure 1 contains only 7 surface glycoproteins and all of them looks to be able to induce neutralizing antibodies, which is not consistent with line 43.
Response 4: We sincerely thank you for your valuable and insightful comments. We have carefully addressed the concerns raised regarding the number of glycoproteins in the PRV envelope and their roles in inducing neutralizing antibodies.
Missing gN Glycoprotein: You are correct that gN was missing from our initial list of glycoproteins. We have now included gN in the revised manuscript on page 1, line 42. Thank you for bringing this oversight to our attention.
Figure 1 Clarification: Our intention was to focus specifically on gB and gE due to their critical roles in viral attachment and entry. While not all glycoproteins were shown in Figure 1, we have clarified in the text that gB, gC, and gD play the most significant roles in inducing neutralizing antibodies, as supported by literature (e.g., Pomeranz et al., 2005; Li et al., 2017). Other glycoproteins, like gE, gI, and gM, are involved in immunoevasion and cell-to-cell spread, but do not elicit as strong a neutralizing antibody response.
We have revised the sentence on page 2, lines 46-47, to ensure consistency with Figure 1 and have provided additional explanation where necessary. Once again, we appreciate your thorough review, which has helped us improve and clarify key aspects of our work. We trust that the revised manuscript now addresses the concerns raised.
Comment 5: Line 44: Please make sure all the descriptions are correct. As gE can be deleted from virus so it’s not a crucial structural component of PRV.
Response 5: Thank you very much for your insightful feedback. You are correct that gE can be deleted from the virus without affecting replication, and it may not be considered a "crucial structural component." However, gE plays a key role in PRV pathogenesis, particularly in immune evasion, cell-to-cell spread, and transmission, especially in neural tissues.
In light of your suggestion, we have revised the wording on page 2, line 48, to reflect that gE is crucial for viral pathogenesis rather than structural integrity. This change aligns with its importance in viral spread and immune evasion, particularly in differentiating between vaccinated and wild-type PRV-infected animals.
Thank you again for helping us improve the accuracy and clarity of our manuscript.
Comment 6: Table 1: Please replace “N Ab” by Protective efficacy as “N Ab levels” are not informative to readers.
Response 6: Thank you for your valuable feedback. I have replaced "N Ab levels" with "Protective efficacy" in Table 1 as suggested.
Comment 7: Please avoid the meaningless sentences such as “PRV gE is used to develop antibodies, as indicated by the study”.
Please read https://www.editage.com/insights/10-tips-to-reduce-the-length-of-your-research-paper/
Response 7: Thank you for your feedback and for providing the helpful reference. We have removed the unnecessary sentences as suggested.
Comment 8: Line 604: “Conclusion”: It is not clear what the authors want to say. In line 608, you mentioned “apply one or more PRV glycoproteins” for novel vaccines then in line 616, you mentioned gE-deleted PRV strains. Which one do you want to recommend and what happed to diagnostic meaning for gE protein?
Response 8: Thank you very much for your insightful comments and for highlighting the need for clarification. We have revised the conclusion section on line 475 of page 13 to clearly outline our recommendations regarding the use of PRV glycoproteins in vaccines and diagnostics. Specifically, we have clarified the application of PRV glycoproteins for vaccine development and addressed the diagnostic significance of gE, as well as the implications of using gE-deleted PRV strains.
Thank you again for your thoughtful review.
Comment 9: I suggest to focus vaccine development as the diagnostic side seems less problematic.
Response 9: Thank you very much for your thoughtful suggestion. We sincerely appreciate your recommendation to focus on vaccine development, as we agree that it is a critical area for addressing the challenges posed by PRV.
However, while vaccine development is indeed essential, we believe the diagnostic aspect also requires attention. Emerging variant strains of PRV pose diagnostic challenges, particularly in differentiating between vaccinated and infected animals. As accurate diagnosis is key to effective PRV control and eradication programs, we feel that balancing both vaccine development and diagnostic improvements is necessary to achieve comprehensive control over the disease.
That said, we value your perspective, and your suggestion has certainly helped us refine the focus of our manuscript. We will ensure that the importance of vaccine development is emphasized while still addressing the diagnostic challenges as part of a holistic approach.
Thank you once again for your valuable input.
Comment 10. Please make sure all the references are properly cited. References such as 4, 6, 7, 22, etc are not complete.
Response 10: We sincerely appreciate your thorough review and valuable feedback. We have carefully reviewed and ensured that all references, including 4, 6, 7, 22, and others, are now correctly cited and complete. Thank you for bringing this to our attention, and we truly value your input in improving the quality of our manuscript.
Minor points:
Minor comment 1. Please provide a Table for abbreviations such as US, UL, TK, DIVA etc.
Response 1: Thank you for your valuable suggestion to include a table for abbreviations such as US, UL, TK, and DIVA. I accepted the comment and provided the list of abbreviations and attached.
Minor comment 2. Line 502: Where is “In this instance”?
Response 2: Thank you for your observation regarding the phrase “In this instance.” I have removed the phrase and revised the paragraph on page 9, starting from line 319, accordingly.

Reviewer 4 Report
Comments and Suggestions for Authors
This is a good review on an important topic. Data were well summarized and discussed. I would suggest minor revision for this manuscript.
1. Please try to cite the most recent published paper in this review. A lot paper cited in this review were published in 1990s.
Author Response
Dear Reviewer 4,
Thank you for your kind and encouraging comments regarding our manuscript. We are pleased to hear that you found the review of the topic important and that the data were well summarized and discussed.
We have carefully considered your suggestions for minor revisions and made the necessary adjustments to improve the manuscript. We believe these changes have enhanced the clarity and overall quality of the work, and we greatly appreciate your insightful guidance throughout this process.
Thank you again for your valuable input and support.
Response to Specific Comment:
Comments 1: Please try to cite the most recent published paper in this review. A lot paper cited in this review were published in 1990s.
Response 1:
Thank you for your thoughtful suggestion regarding the citations. We have revised the manuscript by updating most of the older references from the 1990s with more recent publications where possible.
Where older references were cited, we have replaced them with current publications where applicable. Additionally, we have removed redundant older citations, while keeping both older and newer references where necessary to provide a balanced view. In cases where no recent publications exist and the older references remain essential, we have retained those citations.
We appreciate your insightful feedback, which has helped to strengthen our review.
Round 2
Reviewer 1 Report
Comments and Suggestions for Authors
The authors have responded to the comments in a satisfactorily manner.
Reviewer 3 Report
Comments and Suggestions for Authors
Thank you to the authors for their responses. My concerns have been adequately addressed.